# Optimal Conversion of Conventional Artificial Neural Networks to Spiking Neural Networks

**Shikuang Deng[1] & Shi Gu[1]✉**
School of Computer Science and Engineering
University of Electronic Science and Technology of China
dengsk119@std.uestc.edu.cn, gus@uestc.edu.cn

## Abstract

Spiking neural networks (SNNs) are biology-inspired artificial neural networks (ANNs) that comprise of spiking neurons to process asynchronous discrete signals. While more efficient in power consumption and inference speed on the neuromorphic hardware, SNNs are usually difficult to train directly from scratch with spikes due to the discreteness. As an alternative, many efforts have been devoted to converting conventional ANNs into SNNs by copying the weights from ANNs and adjusting the spiking threshold potential of neurons in SNNs. Researchers have designed new SNN architectures and conversion algorithms to diminish the conversion error. However, an effective conversion should address the difference between the SNN and ANN architectures with an efficient approximation of the loss function, which is missing in the field. In this work, we analyze the conversion error by recursive reduction to layer-wise summation and propose a novel strategic pipeline that transfers the weights to the target SNN by combining threshold balance and soft-reset mechanisms. This pipeline enables almost no accuracy loss between the converted SNNs and conventional ANNs with only $\sim 1/10$ of the typical SNN simulation time. Our method is promising to get implanted onto embedded platforms with better support of SNNs with limited energy and memory. Codes are available at https://github.com/Jackn0/snn_optimal_conversion_pipeline.

## 1 Introduction

Spiking neural networks (SNNs) are proposed to imitate the biological neural networks (Hodgkin & Huxley, 1952a; McCulloch & Pitts, 1943) with artificial neural models that simulate biological neuron activity, such as Hodgkin-Huxley (Hodgkin & Huxley, 1952b), Izhikevich (Izhikevich, 2003), and Resonate-and-Fire (Izhikevich, 2001) models. The most widely used neuron model for SNN is the Integrate-and-Fire (IF) model (Barbi et al., 2003; Liu & Wang, 2001), where a neuron in the network emits a spike only when the accumulated input exceeds the threshold voltage. This setting makes SNNs more similar to biological neural networks.

The past two decades have witnessed the success of conventional artificial neural networks (named as ANNs for the ease of comparison with SNNs), especially with the development of convolutional neural networks including AlexNet (Krizhevsky et al., 2012), VGG (Simonyan & Zisserman, 2014) and ResNet (He et al., 2016). However, this success highly depends on the digital transmission of information in high precision and requires a large amount of energy and memory. So the traditional ANNs are infeasible to deploy onto embedded platforms with limited energy and memory.Distinct from conventional ANNs, SNNs are event-driven with spiking signals, thus more efficient in the energy and memory consumption on embedded platforms (Roy et al., 2019). By far, SNNs have been implemented for image (Acciarito et al., 2017; Diehl & Cook, 2014; Yousefzadeh et al., 2017) and voice (Pei et al., 2019) recognition.

---

✉ Corresponding author

Although potentially more efficient, current SNNs have their own intrinsic disadvantages in training due to the discontinuity of spikes. Two promising methods of supervised learning are backpropagation with surrogate gradient and weight conversion from ANNs. The first routine implants ANNs onto SNN platforms by realizing the surrogate gradient with the customized activation function (Wu et al., 2018). This method can train SNNs with close or even better performance than the conventional ANNs on some small and moderate datasets (Shrestha & Orchard, 2018; Wu et al., 2019; Zhang & Li, 2020; Thiele et al., 2019). However, the training procedure requires a lot of time and memory and suffers from the difficulty in convergence for large networks such as VGG and ResNet. The second routine is to convert ANNs to SNNs by co-training a source ANN for the target SNN that adopts the IF model and soft-reset mechanism (Rueckauer et al., 2016; Han et al., 2020). When a neuron spikes, its membrane potential will decrease by the amount of threshold voltage instead of turning into the resting potential of a fixed value. A limitation of this mechanism is that it discretizes the numerical input information equally for the neurons on the same layer ignorant of the variation in activation frequencies for different neurons. As a consequence, some neurons are difficult to transmit information with short simulation sequences. Thus the converted SNNs usually require huge simulation length to achieve high accuracy (Deng et al., 2020) due to the trade-off between simulation length and accuracy (Rueckauer et al., 2017). This dilemma can be partially relieved by applying the threshold balance on the channel level Kim et al. (2019) and adjusting threshold values according to the input and output frequencies (Han et al., 2020; Han & Roy, 2020). However, as far as we know, it remains unclear how the gap between ANN and SNN formulates and how the simulation length and voltage threshold affect the conversion loss from layers to the whole network. In addition, the accuracy of converted SNN is not satisfactory when the simulation length is as short as tens.

Compared to the previous methods that focus on either optimizing the conversion process or modifying the SNN structures, we theoretically analyze the conversion error from the perspective of activation values and propose a conversion strategy that directly modifies the ReLU activation function in the source ANN to approximate the spiking frequency in the target SNN based on the constructed error form. Our main contributions are summarized as follows:

- We theoretically analyze the conversion procedure and derive the conversion loss that can be optimized layer-wisely.

- We propose a conversion algorithm that effectively controls the difference of activation values between the source ANN and the target SNN with a much shorter simulation length than existing works.

- We both theoretically and experimentally demonstrate the effectiveness of the proposed algorithm and discuss its potential extension to other problems.

## 2 PRELIMINARIES

Our conversion pipeline exploits the threshold balancing mechanism (Diehl et al., 2015; Sengupta et al., 2018) between ANN and SNN with modified ReLU function on the source ANN to reduce the consequential conversion error (Fig.1 A). The modification on regular ReLU function consists of thresholding the maximum activation and shifting the turning point. The thresholding operation suppresses the excessive activation values so that all neurons could be activated within a shorter simulation time. The shift operation compensates the deficit of the output frequency caused by the floor rounding when converting activation values to output frequencies.

We here introduce common notations used in the current paper. Since the infrastructures of the source ANN and target SNN are the same, we use the same notation when it is unambiguous. For the $l$-th layer, we denote $W_l$ as the weight matrix. The threshold on activation value added to the ReLU function in the source ANN is $y_{th}$ and the threshold voltage of the spiking function in the target SNN is $V_{th}$. The SNN is simulated in $T$ time points, where $\boldsymbol{v}^l(t) = \{v_i^l(t)\}$ is the vector collecting membrane potentials of neurons at time $t$, and $\boldsymbol{\theta}^l = \{\theta_i^l(t)\}$ records the output to the next layer, i.e the post synaptic potential (PSP) released by $l$-th layer to the $(l+1)$-th layer. Suppose that within the whole simulation time $T$, the $l$-th layer receives an average input $\boldsymbol{a}_l$ and an average PSP $\boldsymbol{a}_l'$ from the $(l-1)$-th layer corresponding to the source ANN and target SNN, the forward process

can be described by

$$\begin{aligned}
\boldsymbol{a}_{l+1} &= h(W_l \cdot \boldsymbol{a}_l), \\
\boldsymbol{a}'_{l+1} &= h'(W_l \cdot \boldsymbol{a}'_l),
\end{aligned} \tag{1}$$

where $h_l(\cdot)$ and $h'_l(\cdot)$ denote the activation function for the source ANN and target SNN on the average sense respectively. For the source ANN, the activation function $h(\cdot)$ is identical to the activation function for the single input, which is the threshold ReLU function in this work, i.e.

$$h(x) = \begin{cases} 0, & \text{if } x \leq 0; \\ x, & \text{if } 0 < x < y_{th}; \\ y_{th}, & \text{if } x \geq y_{th}, \end{cases} \tag{2}$$

where $y_{th}$ is the threshold value added to the regular ReLU function. For the target SNN, the activation function $h'(\cdot)$ will be derived in the following section.

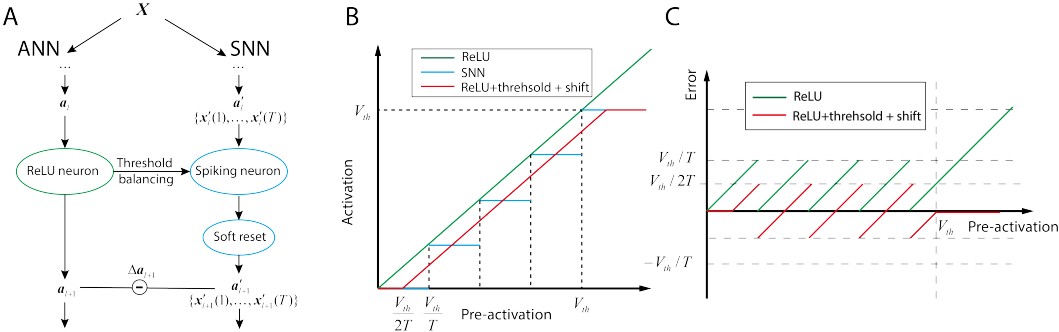

Figure 1: Schematics on the conversion pipeline. (A) The SNN propagates spiking frequencies with the activation sequence $\boldsymbol{x}'_l = \{\boldsymbol{x}'_l(1), ..., \boldsymbol{x}'_l(T)\}$ and averaged output $\boldsymbol{a}'_l$ for $l = 1, ..., L$ through $L$ layers. (B) Activation functions of regular and threshold ReLUs for ANNs, and the step function for SNNs. (C) The error between ReLU and step function with $V_{th}/2T$ shift. See Section 5 for the detailed discussion.

In the following sections, we first derive the equations for weight transform from the source ANN to target SNN. Next, we prove that the overall conversion error can be decomposed to the summation of error between source activation value and target output frequency on each layer. Finally, we estimate an optimal shift value for each layer based on the threshold balancing mechanism (Diehl et al., 2015; Sengupta et al., 2018) and formulate the overall conversion error. Putting these three parts together, we demonstrate both that our approach almost achieves the optimal solution when converting ReLU-based ANNs to spike-based SNNs.

## 3 CONVERSION EQUATION FROM ANN TO SNN

Following the threshold balancing mechanism (Diehl et al., 2015; Sengupta et al., 2018) that copies the weight from the source ANN to the target SNN, we here derive the forward propagation of PSP through layers in the target SNN. For the $l$-th layer in the SNN, suppose it receives its $(t+1)$-th input $\boldsymbol{x}'_l(t+1)$ at time point $t+1$. Then there will be an additional membrane potential $W_l \cdot \boldsymbol{x}'_l(t+1) + \boldsymbol{b}'_l$ added to its membrane potential $\boldsymbol{v}^l(t)$ at time point $t$, resulting in a temporal potential

$$\boldsymbol{v}^l_{temp}(t+1) = \boldsymbol{v}^l(t) + W_l \cdot \boldsymbol{x}'_l(t+1). \tag{3}$$

If any element in $\boldsymbol{v}^l_{temp}(t+1)$ exceeds $V_{th}$, it would release a spike with potential $V_{th}$, decrease its membrane potential by $V_{th}$ and update the membrane potential to $\boldsymbol{v}^l(t+1)$ at time point $t+1$. Thus we can write the membrane potential update rule as:

$$\boldsymbol{v}^l(t+1) = \boldsymbol{v}^l(t) + W_l \cdot \boldsymbol{x}'_l(t+1) - \boldsymbol{\theta}^l(t+1), \tag{4}$$

where $\boldsymbol{\theta}^l(t+1)$ equals $V_{th}$ to release PSP to the next layer if $\boldsymbol{v}_{temp}^l(t+1) \geq V_{th}$ and remain as 0 if $\boldsymbol{v}_{temp}^l(t+1) < V_{th}$. We accumulate the Eqn.4 from time 1 to $T$, divide both sides of the equation by $T$, and have

$$\frac{\boldsymbol{v}^l(T)}{T} = \frac{\boldsymbol{v}^l(0)}{T} + W_l \cdot \sum_{t=1}^{T} \frac{\boldsymbol{x}_l'(t)}{T} - \sum_{t=1}^{T} \frac{\boldsymbol{\theta}^l(t)}{T}. \tag{5}$$

We use $\boldsymbol{a}_l' = \sum_{t=1}^{T} \boldsymbol{x}_l'(t)/T$ to denote the averaged input to the $l$-th layer. Then Eqn5 gives that the input to the $(l+1)$-th layer equals the expected PSP of the $l$-th layer, i.e. $\boldsymbol{a'}_{l+1} = \sum_{t=1}^{T} \boldsymbol{\theta}^l(t)/T$. If we set the initial membrane potential $\boldsymbol{v}^l(0)$ as zero, we can reformulate the Eqn.5 as:

$$\boldsymbol{a}_{l+1}' = W_l \cdot \boldsymbol{a}_l' - \frac{\boldsymbol{v}^l(T)}{T}. \tag{6}$$

When the threshold potential $V_{th}$ is set greater than the maximum of the source ANN's activation values, the remained potential $\boldsymbol{v}^l(T)$ would be less than $V_{th}$ thus would not be output finally. With the clip-operation, the output then can be expressed as

$$\boldsymbol{a}_{l+1}' := h'(W_l \cdot \boldsymbol{a}_l') = \frac{V_{th}}{T} \cdot \text{clip}\left( \left\lfloor \frac{W_l \cdot \boldsymbol{a}_l'}{V_{th}/T} \right\rfloor, 0, T \right), \tag{7}$$

where $\text{clip}(x, 0, T) = 0$ when $x \leq 0$; $\text{clip}(x, 0, T) = x$ when $0 < x < T$; and $\text{clip}(x, 0, T) = T$ when $x \geq T$. So the output function of SNN is actually a step function (see the blue curve in Fig.1 B). The clipping by Eqn.7 is accurate in most cases but can be slightly different from Eqn.6 when the summation of membrane potential cancel between the positive and negative parts while the positive parts get spiked along the sequence. The forward equation for ANN is shown as the green line in Fig.1 B. Since the SNN output is discrete and in the form of floor rounding while the ANN output is continuous, there actually would be an intrinsic difference in $\boldsymbol{a}_{l+1}'$ and $\boldsymbol{a}_{l+1}$ as shown in Fig.1 B,C even if we equalize $\boldsymbol{a}_l'$ and $\boldsymbol{a}_l$. We will analyze how to minimize this difference later.

## 4 DECOMPOSITION OF CONVERSION ERROR

The performance of the converted SNN is determined by the source ANN performance and the conversion error. While the former one is isolated from the conversion, we discuss how to optimize the latter one here. The loss function $\mathcal{L}$ can also be viewed as a function of the last layer output thus the conversion error can be formulated as

$$\Delta \mathcal{L} := \mathbb{E}[\mathcal{L}(\boldsymbol{a}_L')] - \mathbb{E}[\mathcal{L}(\boldsymbol{a}_L)], \tag{8}$$

where the expectation is taken over the sample space. For the $l$-th layer in the SNN, we can reversely approximate its output with the source ANN's activation function $h(\cdot)$ and get

$$\boldsymbol{a}_l' = h_l'(W_l \cdot \boldsymbol{a}_{l-1}') = h_l(W_l \cdot \boldsymbol{a}_{l-1}') + \Delta\boldsymbol{a}_l', \tag{9}$$

where $\Delta\boldsymbol{a}_l'$ is the error caused by the difference between the activation functions of the ANN and SNN. The total output error $\Delta a_l$ between the source ANN and target SNN on the $l$-th layer, i.e. the difference between the activation $\boldsymbol{a}_l$ and $\boldsymbol{a}_l'$ can be approximated as

$$\Delta\boldsymbol{a}_l := \boldsymbol{a}_l' - \boldsymbol{a}_l = \Delta\boldsymbol{a}_l' + [h_l(W_l \cdot \boldsymbol{a}_{l-1}') - h_l(W_l \cdot \boldsymbol{a}_{l-1})] \approx \Delta\boldsymbol{a}_l' + B_l \cdot W_l \cdot \Delta\boldsymbol{a}_{l-1}, \tag{10}$$

where the last approximation is given by the first order Taylor's expansion, and $B_l$ is the matrix with the first derivatives of $h_l$ on the diagonal. Expand the loss function in Eqn.8 around $a_L$, we get

$$\Delta\mathcal{L} \approx \mathbb{E}\left[\nabla_{\boldsymbol{a}_L}\mathcal{L} \cdot \Delta\boldsymbol{a}_L\right] + \frac{1}{2}\mathbb{E}\left[\Delta\boldsymbol{a}_L{}^T H_{\boldsymbol{a}_L} \Delta\boldsymbol{a}_L\right], \tag{11}$$

where the last approximation is given by the 2-order Taylor's expansion, and $H_{a_L}$ is the Hessian of $\mathcal{L}$ w.r.t $a_L$. As $\mathcal{L}$ is optimized on the source ANN, we can ignore the first term here. Thus it suffices to find $\Delta a_L$ to minimize the second term. By substituting Eqn. 10 into Eqn11, we further have

$$\begin{aligned} \mathbb{E}[\Delta\boldsymbol{a}_L{}^T H_{\boldsymbol{a}_L} \Delta\boldsymbol{a}_L] = &\mathbb{E}[\Delta\boldsymbol{a}_L'^T H_{\boldsymbol{a}_L} \Delta\boldsymbol{a}_L'] + \mathbb{E}[\Delta\boldsymbol{a}_{L-1}{}^T B_L W_L^T H_{\boldsymbol{a}_L} W_L B_L \Delta\boldsymbol{a}_{L-1}] \\ &+ 2\mathbb{E}[\Delta\boldsymbol{a}_{L-1}{}^T B_L W_L^T H_{\boldsymbol{a}_L} \Delta\boldsymbol{a}_L'], \end{aligned} \tag{12}$$

where the interaction term can either be ignored by decoupling assumption as in (Nagel et al., 2020) or dominated by the sum of the other two terms by Cauchy's inequality. Applying similar derivation as in (Botev et al., 2017), we have $H_{\boldsymbol{a}_{l-1}} = B_l W_l^T H_{\boldsymbol{a}_l} W_l B_l$. Then Eqn.12 can reduce to

$$
\begin{aligned}
\mathbb{E}[\Delta \boldsymbol{a}_L{}^T H_{\boldsymbol{a}_L} \Delta \boldsymbol{a}_L] &\approx \mathbb{E}[\Delta \boldsymbol{a}_L'^T H_{\boldsymbol{a}_L} \Delta \boldsymbol{a}_L'] + \mathbb{E}[\Delta a_{L-1}{}^T H_{\boldsymbol{a}_{L-1}} \Delta \boldsymbol{a}_{L-1}] \\
&= \sum_l \mathbb{E}[\Delta \boldsymbol{a}_l'^T H_{\boldsymbol{a}_l} \Delta \boldsymbol{a}_l'].
\end{aligned}
\tag{13}
$$

Here the term $H_{\boldsymbol{a}_l}$ can either be approximated with the Fisher Information Matrix (Liang et al., 2019) or similarly assumed as a constant as in (Nagel et al., 2020). For simplicity, we take it as a constant and problem of minimizing the conversion error then reduce to the problem of minimizing the difference of activation values for each layer.

## 5 LAYER-WISE AND TOTAL CONVERSION ERROR

In the previous section, we analyze that minimizing the conversion error is equivalent to minimizing output errors caused by different activation functions of the ANN and SNN on each layer. Here we further analyze how to modify the activation function so that the layer-wise error can be minimized.

First, we consider two extreme cases where (1) the threshold $V_{th}$ is so large that the simulation time $T$ is not long enough for the neurons to fire a spike or (2) the threshold $V_{th}$ is so small that the neuron spikes every time and accumulates very large membrane potential after simulation (upper bound of Eqn.7). For these two cases, the remaining potential contains most of the information from ANN and it is almost impossible to convert from the source ANN to the target SNN. To eliminate these two cases, we apply the threshold ReLU instead of the regular ReLU and set the threshold voltage $V_{th}$ in the SNN as the threshold $y_{th}$ for ReLU in the ANN.

Next, we further consider how to minimize the layer-wise squared difference $\mathbb{E}\|\Delta \boldsymbol{a}_l'\|^2 = \mathbb{E}\|h_l'(W \boldsymbol{a}_{l-1}') - h_l(W \boldsymbol{a}_{l-1}')\|^2$. For the case of using threshold ReLU for $h_l$, the curve of $h_l$ and $h_l'$ are shown in Fig.1 B with their difference in Fig.1 C. We can see that when the input is equal, $h_l$ and $h_l'$ actually have a systematic bias that can be further optimized by either shifting $h_l$ or $h_l'$. Suppose that we fix $h_l'$ and shift $h_l$ by $\delta$, the expected squared difference would be

$$
\mathbb{E}_{\boldsymbol{z}}[h_l'(\boldsymbol{z} - \delta) - h_l(\boldsymbol{z})]^2.
\tag{14}
$$

If we assume that $\boldsymbol{z} = W \boldsymbol{a}_{l-1}'$ is uniformly distributed within intervals $[(t-1)V_{th}/T, tV_{th}/T]$ for $t = 1, ..., T$, optimization of the loss in Eqn.14 can then approximately be reduced to

$$
\arg\min_\delta \frac{T}{2} \cdot \left[ \left( \frac{V_{th}}{T} - \delta \right)^2 + \delta^2 \right] \Rightarrow \delta = \frac{V_{th}}{2T}.
\tag{15}
$$

Thus the total conversion error can be approximately estimated as

$$
\Delta \mathcal{L}_{\min} \approx \frac{L V_{th}^2}{4T},
\tag{16}
$$

which explicitly indicates how the low threshold value and long simulation time decrease the conversion error. In practice, the optimal shift may be different from $V_{th}/2T$. As we illustrate above, the effect of shift is to minimize the difference between the output of the source ANN and the target SNN rather than optimizing the accuracy of SNN directly. Thus the optimal shift is affected by both the distribution of activation values and the level of overfitting in the source ANN and target SNN.

The conversion pipeline is summarized in Algorithm 1 where the source ANN is pre-trained with threshold ReLU. The threshold voltage $V_{th}$ can also be calculated along the training procedure.

## 6 RELATED WORK

Cao et al. (2015) first propose to convert ANNs with the ReLU activation function to SNNs. This work achieves good results on simple datasets but can not scale to large networks on complex data sets. Following Cao et al. (2015), the weight-normalization method is proposed to convert a

---

**Algorithm 1** Conversion from the Source ANN to the Target SNN with Shared Weights

---

**Require:** Pre-trained source ANN, training set, target SNN's simulation length $T$.
**Ensure:** The converted SNN approximates the performance of source ANN with an ignorable error.
  1: Initial $V_{th}^l = 0$, for $l = 1, \cdots, L$ to save the threshold value for each SNN layer.
  2: **for** $s = 1$ to # of samples **do**
  3:     $\mathbf{a}_l \leftarrow$ layer-wise activation value
  4:     **for** $l = 1$ to $L$ **do**
  5:         $V_{th}^l = \max[V_{th}^l, \max(\mathbf{a}_l)]$
  6:     **end for**
  7: **end for**
  8: **for** $l = 1$ to $L$ **do**
  9:     SNN.layer$[l].V_{th} \leftarrow V_{th}^l$
 10:     SNN.layer$[l]$.weight $\leftarrow$ ANN.layer$[l]$.weight
 11:     SNN.layer$[l]$.bias $\leftarrow$ ANN.layer$[l]$.bias $+V_{th}^l/2T$
 12: **end for**

---

three-layer CNN structure without bias to SNN (Diehl et al., 2015; 2016). The spike subtraction mechanism (Rueckauer et al., 2016), also called soft-reset (Han et al., 2020), is proposed to eliminate the information loss caused by potential resetting. More recently, Sengupta et al. (2018) use SPiKE-NORM to obtain deep SNNs like VGG-16 and ResNet-20, and Kim et al. (2019) utilize SNN for object recognition.

The most commonly used conversion method is threshold balancing (Sengupta et al., 2018), which is equivalent to weight normalization (Diehl et al., 2015; Rueckauer et al., 2016). The neurons in the source ANN are directly converted into those in the target SNN without changing their weights and biases. The threshold voltage of each layer in the target SNN is the maximum output of the corresponding layers in the source ANN. The soft reset mechanism (Rueckauer et al., 2017; Han et al., 2020) is effective in reducing the information loss by replacing the reset potential $V_{rest}$ in the hard reset mechanism with the difference between membrane potential $V$ and threshold voltage $V_{th}$ so that the remaining potential is still informative of the activation values. Another big issue for the direct conversion is the varied range of activation for different neurons, which results in a very long simulation time to activate those neurons with high activation threshold values. Rueckauer et al. (2017) suggests using the $p$-th largest outputs instead of the largest one as the weights normalization scale to shrink the ranges. Kim et al. (2019) applies threshold-balance on the channel level rather than the layer level for better adaption. Rathi et al. (2019) initializes the converted SNNs with roughly trained ANNs to shorten simulation length by further fine-training the SNN with STDP and backpropagation. Han et al. (2020), Han & Roy (2020) adjust the threshold and weight scaling factor adapted to the input and output spike frequencies.

## 7 EXPERIMENTS

In this section, we validate our derivation above and compare our methods with existing approaches via converting CIFAR-Net, VGG-16, and ResNet-20 on CIFAR-10, CIFAR-100, and ImageNet. See Appendix A.1 for the details of the network infrastructure and training parameters.

### 7.1 ANN PERFORMANCE WITH THRESHOLD RELU

We first evaluate the impact of modifying ReLU on the source ANN from the perspective of maximum activation value distribution and classification accuracy. A smaller range of activation values will benefit the conversion and but may potentially result in lower representativity for the ANN.

We set the threshold value $y_{th} = 1$ on CIFAR-10, $y_{th} = 2$ on CIFAR-100 and examine the impact of threshold on the distribution of activation values. From Fig.2A, we can see that the threshold operation significantly reduces the variation of maximum activation values across layers and the shift-operation won't cause any additional impact significantly. This suggests that with the modified ReLU, the threshold voltage $V_{th}$ of spiking could be more effective to activate different layers. We next look into the classification accuracy when modifying the ReLU with threshold and shift. Comparisons on CIFAR-10 and CIFAR-100 are shown in Fig.2B. On the CIFAR-10, the perfor-

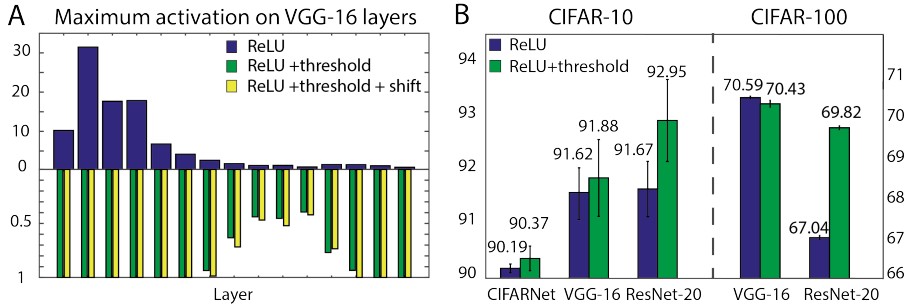

Figure 2: Impact of threshold on ANN. (A)Maximum activations of VGG-16's layers on CIFAR-10. (B) The accuracy of ANN with regular or threshold ReLU on different networks. Average over 5 repeats on CIFAR-10 and 3 repeats on CIFAR-100.

mance of networks with modified ReLU actually get enhanced for all three infrastructures ($+0.18\%$ on CIFAR-Net, $+0.26\%$ on VGG-16 and $+1.28\%$ on ResNet-20). On the CIFAR-100 dataset, the method of adding threshold results in a slight decrease in accuracy on VGG-16 ($-0.16\%$), but a huge increase on ResNet-20 ($+2.78\%$). These results support that the threshold ReLU can serve as a reasonable source for the ANN to SNN conversion.

## 7.2 IMPACT OF SIMULATION LENGTH AND SHIFT ON CLASSIFICATION ACCURACY

In this part, we further explore whether adding threshold and shift to the activation function will affect the simulation length ($T$) required for the converted SNN to match the source ANN in classification accuracy and how shift operation affects differently in different periods of simulation.

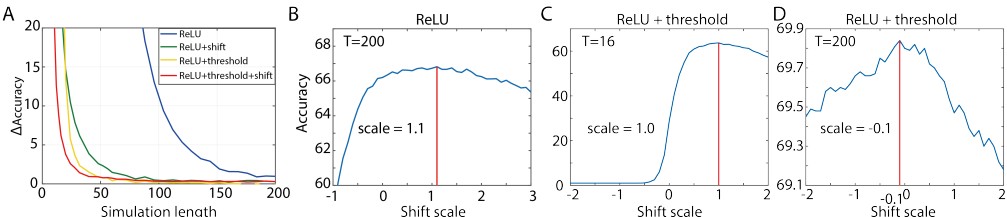

Figure 3: Impact of threshold and shift on convergence for converting ResNet-20 on CIFAR-100. (A) SNN's accuracy losses w.r.t different simulation lengths. (B-D) Optimal shifts w.r.t different activation functions and simulation lengths.

From Fig.3A, we can see that no matter applied separately or together, the threshold and shift operations always significantly shorten the simulation length to less than 100 when the accuracy loss is much smaller than using regular ReLU with $T \approx 200$. When $T < 50$, the combination of threshold and shift achieves the fastest convergence and the adoption of threshold is more efficient than shift. However, the difference is minor when simulating with $T > 100$. This inspires us to further explore the impact of shift at different simulation length. From the green and blue lines in Fig.2A and accuracy changes w.r.t the variation of shift scales in Fig.2B, we can conclude that the shift by $V_{th}/2T$ is almost always optimal even when $T = 200$ for the source ANN with regular ReLU. However, the shift works differently when the ReLU is with threshold. When the simulation length is very short like $T = 16$, the conversion can benefit hugely ($> 0.30$ accuracy improvement) from the derived optimal shift (see Fig.2C). When the simulation length is long enough, the shift mechanism no longer works and may slightly damage the accuracy (Fig.2D) while adding threshold alone will almost eliminate the conversion error that consistently remains above zero in the original threshold balancing approach (Diehl et al., 2015; Sengupta et al., 2018) as shown in Appendix Fig.4.

### 7.3 COMPARISON WITH RELATED WORK

In order to validate the effectiveness of the whole proposed pipeline, here we compare our method with SPIKE-NORM (Sengupta et al., 2018), Hybrid Training (Rathi et al., 2019), RMP (Han et al., 2020), TSC(Han & Roy, 2020) and direct training method including STBP (Wu et al., 2019) and TSSL-BP (Zhang & Li, 2020) through classification tasks on CIFAR-10, CIFAR-100 and ImageNet (Table.1). For the network with relatively light infrastructure like CIFAR-Net, our model can converge to $> 90\%$ accuracy within 16 simulation steps. Direct training methods like STBP and TSSL-BP are feasible to achieve a high accuracy with short simulation length. However, their training cost is high due to RNN-like manner in the training procedure and cannot extend to complex network infrastructures like VGG-16 and ResNet-20. RMP and TSC achieve the highest accuracy for SNN with VGG-16, probably a result from more accurate pre-trained source ANNs. For ResNet-20, our model outperforms the compared models not only in accuracy but also in the simulation length.

Table 1: Comparison between our work and other conversion methods. The conversion loss is reported as the accuracy difference ($acc_{ANN} - acc_{SNN}$) between the source ANN with regular ReLU and threshold ReLU (in brackets) and the converted SNN. Ourwork-X-TS denotes the SNN converted from the source ANN with both threshold (T) ReLU and shift (S).

| Method | CIFAR-10 + CIFAR-Net | | | CIFAR-10 + VGG-16 | | | CIFAR-10 + ResNet-20 | | |
|---|---|---|---|---|---|---|---|---|---|
| | Accuracy | Conversion Loss | Length | Accuracy | Conversion Loss | Length | Accuracy | Conversion Loss | Length |
| STBP (w/o NeuNorm) | 89.83% | 0.66% | 8 | NA | | | NA | | |
| STBP (w/ NeuNorm) | 90.53% | −0.04% | 8 | NA | | | NA | | |
| TSSL-BP | 91.41% | −0.92% | 5 | NA | | | NA | | |
| SPIKE-NORM | NA | | | 91.55% | 0.15% | 2500 | 87.46% | 1.64% | 2500 |
| RMP | NA | | | 93.63% | 0.01% | 1536 | 91.36% | 0.11% | 2048 |
| Hybrid Training | NA | | | 91.13% | 1.68% | 100 | 92.22% | 0.93% | 250 |
| TSC | NA | | | 93.63% | 0.00% | 2048 | 91.42% | 0.05% | 1536 |
| Our Work-1-TS | 90.22% | 0.06%(0.41%) | 16 | 92.29% | −0.2%(0.05%) | 16 | 92.41% | −0.09%(1.20%) | 16 |
| Our Work-2-TS | 90.46% | −0.18%(0.17%) | 32 | 92.29% | −0.2%(0.05%) | 32 | 93.30% | −0.98%(0.31%) | 32 |
| Our Work-3-TS | 90.58% | −0.20%(0.05%) | 64 | 92.22% | −0.13%(0.12%) | 64 | 93.55% | −1.23%(0.06%) | 64 |
| Our Work-4-TS | 90.58% | −0.20%(0.05%) | 128 | 92.24% | −0.15%(0.10%) | 128 | 93.56% | −1.25%(0.05%) | 128 |
| Our Work-5-T | 90.61% | −0.23%(0.02%) | 400-600 | 92.26% | −0.17%(0.08%) | 400-600 | 93.58% | −1.26%(0.03%) | 400-600 |
| Our Work-6-S | 90.13% | 0.05% | 512 | 92.03% | 0.06% | 512 | 92.14% | 0.18% | 512 |
| Method | CIFAR-100 + VGG-16 | | | CIFAR-100 + ResNet20 | | | ImageNet + VGG-16 | | |
| | Accuracy | Conversion Loss | Length | Accuracy | Conversion Loss | Length | Accuracy | Conversion Loss | Length |
| SPIKE-NORM | 70.77% | 0.45% | 2500 | 64.09% | 4.63% | 2500 | 69.96% | 0.56% | 2500 |
| RMP | 70.93% | 0.29% | 2048 | 67.82% | 0.9% | 2048 | 73.09% | 0.4% | 4096 |
| Hybrid Training | NA | | | NA | | | 65.19% | 4.16% | 250 |
| TSC | 70.97% | 0.25% | 1024 | 68.18% | 0.54% | 2048 | 73.46% | 0.03% | 2560 |
| Our Work-1-TS | 65.94% | 4.68%(4.55%) | 16 | 63.73% | 3.35%(6.07%) | 16 | 55.80% | 16.6%(16.38%) | 16 |
| Our Work-2-TS | 69.80% | 0.82%(0.69%) | 32 | 68.40% | −1.32%(1.40%) | 32 | 67.73% | 4.67%(4.45%) | 32 |
| Our Work-3-TS | 70.35% | 0.27%(0.14%) | 64 | 69.27% | −2.19%(0.53%) | 64 | 70.97% | 1.43%(1.21%) | 64 |
| Our Work-4-TS | 70.47% | 0.15%(0.02%) | 128 | 69.49% | −2.41%(0.31%) | 128 | 71.89% | 0.51%(0.29%) | 128 |
| Our Work-5-T | 70.55% | 0.07%(−0.06%) | 400-600 | 69.82% | −2.74%(−0.02%) | 400-600 | 72.17% | 0.23%(0.01%) | 400-600 |
| Our Work-6-S | 70.28% | 0.31% | 512 | 66.63% | 0.45% | 512 | 72.34% | 0.06% | 512 |

We next pay attention to the accuracy loss during the conversion. For VGG-16, our proposed model averaged on length 400-600 achieves a similar loss with RMP on CIFAR-10 ($\approx 0.01\%$). On CIFAR-100 and ImageNet, our model maintains the lowest loss compared to both the ANN trained with regular ReLU and threshold ReLU. For ResNet-20, there appears an interesting phenomenon that the conversion to SNN actually improves rather than damage the accuracy. Compared with the source ANN trained with regular ReLU, the accuracy of our converted SNN has a dramatic increase with 1.26% on CIFAR-10 and 2.74% on CIFAR-100. This may be a result from the fact that the source ANN for conversion is not compatible with the batch normalization and max pooling thus bears a deficit by overfitting that can be reduced by the threshold and discretization of converting to SNN.

In addition to the excellence in accuracy, our model is even more advantageous in the simulation length. When we combine the threshold ReLU with shift operation through the layer-wise approximation, our model can achieve a comparable performance with RMP for VGG-16 with 16 simulation steps on CIFAR-10 and 32 simulation steps on CIFAR-100, much shorter than that of RMP (1536 on CIFAR-10, 2048 on CIFAR-100). Similar comparisons are observed on ResNet-20 with better performance than RMP, TSC, SPIKE-NORM and Hybrid Training. On ImageNet, although we do not achieve the highest accuracy due to the constraint of accurate source ANN and short simulation length, we can still observe that the 400-600 simulation steps can return a conversion error smaller than that of RMP, TSC and SPIKE-NORM with 4096 and 2500 steps correspondingly. When the simulation length is 512, the conversion with only shift operation works accurately enough as well.

See Appendix A.2 for the repeated tests averaged with simulation length 400 to 600 and Appendix A.3 for the comparison with RMP and TSC when the simulation length is aligned on a short level.

The proposed algorithm can potentially work for the conversion of recurrent neural network (RNN) as well. See Appendix A.4 for an illustrative example that demonstrates the superiority of the current work over directly-training approach when the source architecture is an RNN.

## 8 DISCUSSION

Since Cao et al. (2015) first proposed conversion from ANN with ReLU to SNN, there have been many follow-up works (Diehl et al., 2016; Rueckauer et al., 2016; Sengupta et al., 2018; Han et al., 2020) to optimize the conversion procedure. But they fail to theoretically analyze the conversion error for the whole network and neglect to ask what types of ANNs are easier to transform. We fill this gap and performed a series of experiments to prove their effectiveness.

Our theoretical decomposition of conversion error actually allows a new perspective of viewing the conversion problem from ANNs to SNNs. Indeed, the way we decompose the error on activation values works for the problem of network quantization (Nagel et al., 2020) as well. This is because the perturbation in edge weight, activation function and values would all affect the following layers through the output values of the current layer. However, one limitation to point out here is that the interaction term in Eqn.12 may not be small enough to ignore when different layers are strongly coupled in the infrastructure. In this case, the layer-wise recursion is suboptimal and blockwise approaches may help for further optimization. The trade-off between representativity and generalizability applies for the adoption of threshold ReLU in the training of ANN and provides an intuitive interpretation on why our converted SNN outperforms the source ResNet-20 in the conversion.

We also show that shift operation is necessary for the conversion considering the difference between ReLU and step function. This difference has also been noticed in direct training approach as well. Wu et al. (2018; 2019) use a modified sign function to model the surrogate gradient. This choice makes the surrogate curve coincide with the red line in Fig.2B. Yet, we need to point out that the shift operation cannot ensure the improvement when the simulation length is long enough. We can understand the shift operation as an approach to pull the target SNN to the source ANN. At the same time, pulling the SNN to ANN also causes it to reduce the generalizability out of discretization. As we describe above, the target SNN is possible to outperform the source ANN by benefiting more generalizability than losing the representativity. In this case, it makes no sense to pull the target SNN model back to the source ANN anymore. This is potentially the reason why the optimal shift is no longer $V_{th}/2T$ when the simulation length is long enough to make the target SNN hardly gain more representativity than lose generalizability by getting closer to the source ANN.

Compared to the direct training approach of SNNs, one big limitation of converting SNNs from ANNs is the simulation length. Although SNNs can reduce the energy cost in the forward process (Deng et al., 2020), the huge simulation length will limit this advantage. Our method greatly reduces the simulation length required by the converted SNNs, especially for large networks and can be combined with other optimization methods in Spiking-YOLO, SPIKE-NORM and RMP. Further, as we point above about the duality between input $\mathbf{x}$ and edge weight $\mathbf{w}$, future optimization could potentially reduce the simulation length to about $2^4 = 16$ that is equivalent to 4 bits in network quantization with very tiny accuracy loss.

In the future, it makes the framework more practical if we can extend it to converting source ANN trained with batch normalization and activation functions taking both positive and negative values.

## 9 ACKNOWLEDGMENT

This project is primarily supported by NSFC 61876032.

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

## A Appendix

### A.1 Network structure and Optimization Setup

We adopt three network structures, CIFARNet (Wu et al., 2019), VGG-16 and ResNet-20 (Sengupta et al., 2018). All the pooling layers use average pooling to adapt to SNN. CIFARNet is a 8 layer structure like: 128C3 (Encoding)-256C3-AP2-512C3-AP2-1024C3-512C3-1024FC-512FC-Voting, where C means convolutional layer, AP means average pooling and FC means fully connected layer. We use the VGG-16 and ResNet-20 model provided by Rathi et al. (2019) on the CIFAR dataset and standard VGG-16 on ImageNet. ResNet-20 is ResNet-18 adding two convolutional layers for pre-processing in front, and an activation function is added after the basic blocks (Sengupta et al., 2018). Following (Sengupta et al., 2018), before training, we initialize convolution layers (kernel size k, and n output channels) weight a normal distribution and standard deviation $\sqrt{2/(k^2 n)}$ for non-residual convolutional layer, $\sqrt{2}/(k^2 n)$ for residual convolutional layer. And then we add dropout layer that $p = 0.2$ between convolutional layers and $p = 0.5$ between fully connected layers.

Our work is based on Pytorch platform. For CIFAR-10 and CIFAR-100, the initial learning rate of each network is 0.01, batch size is 128, total epochs is 300. We use cross entropy loss function and SGD optimizer with momentum 0.9 and weight decay $5e - 4$. And learning rate decay by a factor of 0.1 at epochs of 180, 240 and 270. For the sake of better conversion, in the first 20 epochs, we use regular ReLU for training, and then add the threshold to ReLU. On CIFAR-100, our network is initialized with the network parameters on CIFAR-10 except the output layer (Pei et al., 2019). The training on CIFAR-100 adopts the pre-trained weight without threshold for initialization in the first few epochs as a compensation of the lack of batch-normalization (Han et al., 2020; Rathi et al., 2019) to enable the convergence in early steps.

### A.2 SNN performance with simulation length 400-600

We provide the average accuracy and variance of the converted SNN over the simulation length of 400-600 on CIFAR-10 (Table2) and CIFAR-100 datasets (Table3), the shift operation is shifting constant $V_{th}/1600$, except for the output layer. The results include (1) regular ReLU without our method, (2) threshold ReLU and (3) both threshold and shift operation. Since simulation length of 400-600 is long enough for conversion from the source ANN with threshold ReLU to SNN, shift operation may reduce the converted SNN accuracy.

Table 2: Performance of the converted SNN on CIFAR-10 (averaged on simulation length 400-600).

| Structure | ReLU | ReLU+threshold | ReLU+threshold+shift |
|---|---|---|---|
| CIFAR-Net | $90.16 \pm 0.03$ | $90.61 \pm 0.01$ | $90.37 \pm 0.02$ |
| VGG-16 | $91.82 \pm 0.09$ | $92.26 \pm 0.01$ | $92.26 \pm 0.01$ |
| ResNet-20 | $92.14 \pm 0.03$ | $93.58 \pm 0.01$ | $93.59 \pm 0.01$ |

Table 3: Performance of the converted SNN on CIFAR-100 (averaged on simulation length 400-600).

| Structure | ReLU | ReLU+threshold | ReLU+threshold+shift |
|---|---|---|---|
| VGG-16 | $70.10 \pm 0.05$ | $70.55 \pm 0.03$ | $70.43 \pm 0.03$ |
| ResNet-20 | $66.74 \pm 0.04$ | $69.82 \pm 0.03$ | $69.81 \pm 0.02$ |

### A.3 SNN performance with short simulation length

Here we compare the performance of our method versus RMP and TSC when the simulation length is short. The conversion loss is reported as the accuracy difference ($acc_{ANN} - acc_{SNN}$) between the converted SNN and the source ANN with regular ReLU and threshold ReLU (in brackets). Our

work-T/S denotes the SNN converted from the source ANN with both threshold (T) ReLU and shift (S). The performance of using shift operation and regular ReLU is similar to TSC on CIFAR-10 (Table4), but is better than TSC and RMP on CIFAR-100 dataset (Table5). For VGG-16 on ImageNet, our model can achieve nearly zero conversion loss with simulation length 256 (Table 6). Using both threshold and shift at the same time reduces the conversion error the fastest when simulation length is short. And only using threshold achieves the minimum conversion error on a relatively long simulation time.

Table 4: Compare the conversion loss with short simulation length for VGG-16 and ResNet-20 on CIFAR-10.

| Method | 32 | 64 | 128 | 256 | 512 |
|---|---|---|---|---|---|
| **VGG-16 on CIFAR-10** | | | | | |
| RMP | 33.33% | 3.28% | 1.22% | 0.59% | 0.24% |
| TSC | NA | 0.84% | 0.36% | 0.18% | 0.06% |
| Our Work-S | 29.71% | 3.10% | 0.99% | 0.60% | 0.06% |
| Our Work-T | 0.06%(0.31%) | $-0.21\%(0.04\%)$ | $-0.18\%(0.07\%)$ | $-0.18\%(0.07\%)$ | $-0.16\%(0.09\%)$ |
| Our Work-TS | $-0.2\%(0.05\%)$ | $-0.13\%(0.12\%)$ | $-0.15\%(0.10\%)$ | $-0.19\%(0.06\%)$ | $-0.16\%(0.09\%)$ |
| **ResNet-20 on CIFAR-10** | | | | | |
| RMP | 16.78% | NA | 3.87% | 2.1% | 0.84% |
| TSC | NA | 22.09% | 2.9% | 1.37% | 0.38% |
| Our Work-S | 4.24% | 1.13% | 0.39% | 0.21% | 0.18% |
| Our Work-T | 19.72%(21.01%) | $-0.98\%(0.31\%)$ | $-1.14\%(0.15\%)$ | $-1.27\%(0.02\%)$ | $-1.28\%(0.01\%)$ |
| Our Work-TS | $-0.98\%(0.31\%)$ | $-1.23\%(0.06\%)$ | $-1.25\%(0.05\%)$ | $-1.25\%(0.05\%)$ | $-1.28\%(0.01\%)$ |

Table 5: Compare the conversion loss with short simulation length for VGG-16 and ResNet-20 on CIFAR-100.

| Method | 32 | 64 | 128 | 256 | 512 |
|---|---|---|---|---|---|
| **VGG-16 on CIFAR-100** | | | | | |
| RMP | NA | NA | 7.46% | 2.88% | 1.82% |
| TSC | NA | NA | 1.36% | 0.57% | 0.35% |
| Our Work-S | NA | NA | 1.13% | 0.55% | 0.31% |
| Our Work-T | NA | NA | 0.21%(0.08%) | $0.12\%(-0.01\%)$ | $0.07\%(-0.06\%)$ |
| Our Work-TS | NA | NA | 0.15%(0.02%) | $0.08\%(-0.05\%)$ | 0.14%(0.01%) |
| **ResNet-20 on CIFAR-100** | | | | | |
| RMP | 41.08% | 21.81% | 11.03% | 4.66% | 2.56% |
| TSC | NA | NA | 10.03% | 3.45% | 1.55% |
| Our Work-S | 6.15% | 1.61% | 0.70% | 0.52% | 0.45% |
| Our Work-T | $-0.14\%(2.58\%)$ | $-2.18\%(0.53\%)$ | $-2.62\%(0.10\%)$ | $-2.78\%(-0.06\%)$ | $-2.76\%(-0.04\%)$ |
| Our Work-TS | $-1.32\%(1.4\%)$ | $-2.19\%(0.53\%)$ | $-2.41\%(0.31\%)$ | $-2.41\%(0.31\%)$ | $-2.54\%(0.18\%)$ |

Table 6: Compare the conversion loss with short simulation length on VGG-16 and ImageNet.

| Method | 256 | 512 |
|---|---|---|
| RMP | 24.56% | 3.95% |
| TSC | 3.75% | 0.87% |
| Our Work-S | 0.41% | 0.06% |
| Our Work-T | 0.35%(0.15%) | 0.22%(0.02%) |
| Our Work-TS | 0.26%(0.06%) | 0.25%(0.05%) |

## A.4 PERFORMANCE OF SNN CONVERTED FROM RNN

Here we provide an illustrative example of how the proposed conversion pipeline can be extended to the case of converting RNN on the dataset for Sentiment Analysis on Movie Reviews (Socher et al., 2013). The rules are slightly different from conversion in the main text considering the

implicit definition of RNN's hidden states. For the fairness of comparison, we set the same input and simulation length for the RNN and SNN and adjust the structure as follows. (1) The source RNN adopts the threshold-ReLU as its activation function with the remaining value as the hidden state value. The hidden value will be added to the pre-activation value at the next time point. (2) We add a non-negative attenuation $\tau$ to the hidden state values in order to enhance the nonlinearity. (3) The converted SNN keeps the same infrastructure, weight parameters, attenuation value $\tau$ as the source RNN. (4) On the output layer, we use the threshold balancing method and loop its input multiple times to fire enough spikes to obtain a good approximation to the fully-connected layer for the final prediction.

We compare the performance of the source RNN, converted SNN and directly-trained SNN. On the validation set, the converted SNN achieves an accuracy of 0.5430 that is close to the source RNN (acc = 0.5428), while the directly-trained SNN with surrogate gradient only gets an 0.5106 accuracy. We also find that using the regular ReLU instead of threshold-ReLU on the source RNN produces a big accuracy drop for the converted SNN (acc = 0.5100). Since the surrogate gradient error of the directly-trained method will accumulate over the whole simulation process while complex infrastructures and tasks usually require a long simulation to achieve an effective spiking frequency distribution, the typical directly-training approaches for SNNs are often not optimal in complex network structures and tasks. Our results illustrate the potential efficiency of converting SNN from RNN. In future works, it would be promising to investigate how to design a better conversion strategy that can simultaneously combine the strength of RNN and SNN.

### A.5 COMPARE THE PERFORMANCE ON LONG SIMULATION

As a supplement to Fig.3 A, here we compare the conversion accuracy loss of the SNNs converted from the source ANNs with regular and threshold ReLU functions along an extended simulation length. Although the conversion loss in both cases decays fairly fast, the SNN converted from the ANN with regular ReLU function converges to a plateau that suffers about 0.75% accuracy loss while the SNN converted from the ANN with threshold ReLU is almost loss-free. This result indicates that the improvement by adding threshold is not only on the converging efficiency but also on the final performance, which cannot be easily compensated through extending the simulation time in the original threshold balancing approach.

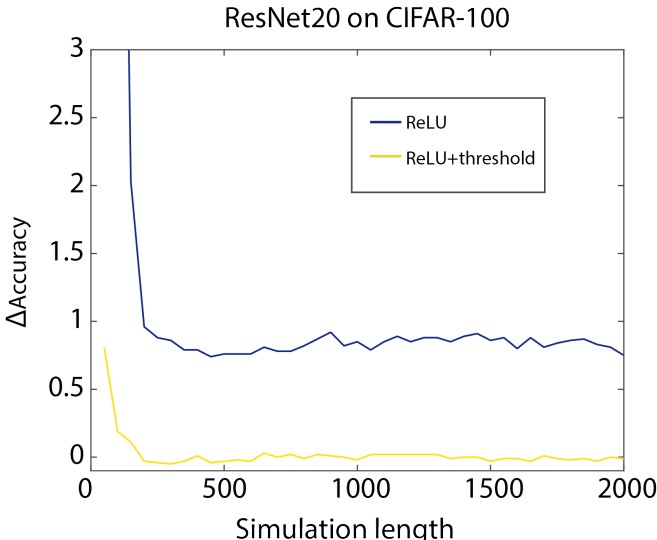

Figure 4: The accuracy gap between the source ANN and target SNN along an extended simulation length.

