# OpenReview forum: "Optimal Conversion of Conventional Artificial Neural Networks to Spiking Neural Networks"
_ICLR.cc/2021/Conference — ICLR 2021 Poster_

### Official Review · AnonReviewer4 · 2020-10-23
**This paper proposes a layer-wise decomposition of the conversion error method to optimize the ANN-SNN conversion.**

**Rating:** 7
**Confidence:** 3

**Review:**

Strength:
(1) This paper proposes a layer-wise optimization method for ANN-SNN conversion. I appreciate the theoretical analysis of how to minimize the conversion error.

(2) This work significantly reduces the simulation time since long simulation time is usually required for converted SNN to reduce error.

(3) It's interesting that conversion to SNN actually improves rather than damage the accuracy on ResNet.

Weakness:
(1) Although the proposed method is much more efficient, it does not show obvious performance improvement compared to existing methods.

(2) In the experiment, the ResNet consistently show better performance. I hope the authors can provide more comments on this.

(3) I'm not sure if I missed anything. The threshold RELU is not defined in the paper which may cause confusion.

(4) I hope the authors can summarize the whole steps by formula or algorithm to help readers understand the entire process.

---

> ### Author Response · Authors · 2020-11-15
> **Response to Review # 4**
>
> We thank the reviewer for his/her time in reviewing our work and appreciate the helpful suggestions to further improve the quality of our work. We’ve made the reversion accordingly in the manuscript and summarize the response below for ease of review.
>
> ##### Response to weakness 1:
> The improvement reported in Table 1 is not very obvious due to the setup of the conversion from ANN to SNN. Within this setup, the source ANN is usually given. When the simulation step T is large enough, the conversion error would go zero, driving the converted SNN to approximate the performance of the source ANN. In the inference step of SNN, an image needs to be processed with T-step simulations. Thus the most critical part of converting ANN to SNN is actually the improvement of efficiency with a shorter simulation length, which is beneficial to not only the training but also the deployment of the SNN network.
>
> We agree with the reviewer that making SNN achieve a higher performance than the traditional ANN would fully release the potential power of SNN. But the direct training of the SNN with complex infrastructure is beyond the scope of the current work and still an open problem in the whole field.
>
> ##### Response to weakness 2:
> The improvement on ResNet is relatively larger because the source ANN for the conversion is indeed trained without the batch-normalization thus its generalizability is a bit damaged compared to the ResNet with batch-normalization. The way we build up the converted SNN with threshold ReLU and shorter simulation length indeed make up for this loss of generalizability by limiting and discretizing the range of activation values. By Liang [1], these approaches could decrease the Fisher-Rao norm that acts as an upper bound measuring the generalizability thus potentially increase the generalizability when we test the model performance on the testing dataset.
>
> The common batch-normalization is not suitable for the converted SNN as it will move the distribution of activation values to the negative range thus destabilize the convergence of the SNN as the spiking frequency can only be positive. So, recent conversion methods all use the ANN trained without batch-normalization. In future works, it will make sense to extend the threshold ReLU to the leaky version and build the SNN with inhibitory layers to generate negative activation values equivalent to the negative frequencies.
>
> ##### Response to weakness 3:
> We appreciate the suggestion of adding an explicit definition of threshold RELU to avoid potential confusion. It is now defined in Section 2 PRELIMINARIES on page 2.
>
> ##### Response to weakness 4:
> We adopt this helpful suggestion to improve the readability and add the description of the conversion process as Algorithm 1 on page 5.
>
> [1] Liang, Tengyuan, Tomaso Poggio, Alexander Rakhlin, and James Stokes. "Fisher-rao metric, geometry, and complexity of neural networks." In The 22nd International Conference on Artificial Intelligence and Statistics, pp. 888-896. 2019.

---

### Official Review · AnonReviewer2 · 2020-10-29
**Training spiking neural nets to copy non-spiking ones**

**Rating:** 7
**Confidence:** 3

**Review:**

The authors seek a mechanism to train a spiking neural net to duplicate the function of a non-spiking one. This is desirable for energy-efficient inference, although the training process becomes challenging due to the discrete nature of the spiking process.

To achieve their goal, they described the spiking neuron non-linearity by a "staircase" function of the input (spiking output increases by 1 each time the input gets big enough to reach the next stair), and related that to the ReLu function used in the non-spiking neural net. They then determined parameters for the modified ReLU that would minimize the deviation between these activation functions, and computed the minimum conversion error (for converting ANN -> SNN). This scales with the square of the threshold voltage for spiking, divided by the simulation time. As one might expect, lower thresholds, and longer simulation times, both of which lead to potentially higher spike counts and thus lower discretization errors, lead to smaller conversion errors.

Using this, they defined their procedure for training SNN to mimic ANN as follows: they trained the ANN with their modified ReLU (which is closer to the SNN activation function but more readily differentiable), and then used the weights from that ANN in their SNN.

Next, the authors evaluated their procedure on several different image categorization networks. Nice performance was obtained in all cases: better than using a normal ReLU, or other comparison activation functions, in the "target" ANN.

Overall, this is a reasonably nice piece of work. I'd like to see this applied to recurrent neural nets: there, the dynamics of the SNN could be used more naturally, and the results might be more meaningful.

---

> ### Author Response · Authors · 2020-11-15
> **Response to Review # 2**
>
> We thank the reviewer for his/her time in reviewing our work and appreciate the helpful suggestions to further improve the quality of our work. The conversion from RNN to SNN is a bit different from the current work as it requires a different set of terminology and the extension of SNN to negative spiking frequencies.  As far as we know, there is NO reported result that systematically talks about the conversion from RNN to SNN. Instead of rigorously building the conversion theory which may be beyond the current scope of the paper, here we show an illustrative example to demonstrate the power of the proposed framework.  We add the results in Appendix A.4 to keep the completeness of the current work. For ease of review, we copy the paragraphs below.
> ______________________________________________________________________________________________________________________________
> ##### Performance of SNN converted from RNN
> Here we provide an illustrative example of how the proposed conversion pipeline can be extended to the case of converting RNN on the dataset for Sentiment Analysis on Movie Reviews \citep{socher2013recursive}. The rules are slightly different from conversion in the main text considering the implicit definition of RNN's hidden states. For the fairness of comparison, we set the same input and simulation length for the RNN and SNN and adjust the structure as follows. (1) The source RNN adopts the threshold-ReLU as its activation function with the remaining value as the hidden state value. The hidden value will be added to the pre-activation value at the next time point. (2) We add a non-negative attenuation $\tau$ to the hidden state values in order to enhance the nonlinearity. (3) The converted SNN keeps the same infrastructure, weight parameters, attenuation value $\tau$ as the source RNN. (4) On the output layer, we use the threshold balancing method and loop its input multiple times to fire enough spikes to obtain a good approximation to the fully-connected layer for the final prediction.
>
> We compare the performance of the source RNN, converted SNN and directly-trained SNN. On the validation set, the converted SNN achieves an accuracy of 0.5430 that is close to the source RNN (acc = 0.5428), while the directly-trained SNN with surrogate gradient only gets an 0.5106 accuracy. We also find that using the regular ReLU instead of threshold-ReLU on the source RNN produces a big accuracy drop for the converted SNN (acc = 0.5100). Since the surrogate gradient error of the directly-trained method will accumulate over the whole simulation process while complex infrastructures and tasks usually require a long simulation to achieve an effective spiking frequency distribution, the typical directly-training approaches for SNNs are often not optimal in complex network structures and tasks. Our results illustrate the potential efficiency of converting SNN from RNN. In future works, it would be promising to investigate how to design a better conversion strategy that can simultaneously combine the strength of RNN and SNN.

---

> > ### Comment · AnonReviewer2 · 2020-11-19
> > **Good addition**
> >
> > I appreciate the addition of the RNN study in the appendix. That topic needs to be fleshed out more -- but in another paper. For the scope of this paper, that's good enough I think. You have shown that the approach is feasible for the RNN case (which to me is the more interesting one), which should prompt follow-on work to study that in more depth.
> >
> > Nice.

---

> > > ### Author Response · Authors · 2020-11-20
> > > **Thanks for AnonReviewer2**
> > >
> > > Thank you for the acknowledgment of our current work and constructive suggestions for future directions. We will explore more on the RNN in the following works.

---

### Official Review · AnonReviewer3 · 2020-10-29
**Interesting (second review)**

**Rating:** 5
**Confidence:** 3

**Review:**

## Edit on second review

I apologize again for the tone of my first review, I sincerely tried to understand the paper but I could not when I first read it. A re-read the paper and finally understood it during the review. I left a comment to the authors in the discussion below and they appropriately addressed my new recommendations. With the new equation (1) the paper is hopefully more understandable now.

I increase my grade from 3 to 5. The findings are quite interesting but I still believe that the paper is not well written: the equations are interesting but the explanations between the equations are often unclear. One has to understand each equation and be quite imaginative to finally identify the contributions of the paper (even for somebody only "very slightly" off from the research topic).

## Summary

The authors suggest a relationship between a leaky relu and a spiking integrate and firing neuron model.
This relationship suggests a mapping between the two models which is imperfect, a loss seems to be derived to reduce this mismatch along the network training. The method is tested on CIFAR-10 and CIFAR-100, and compared with some other methods for converting ANNs to SNNs.

## Critical review

This topic is potentially important since spiking neural network are gaining popularity. But this paper is clearly badly written and it is extremely hard to understand, both in the math and in the text. I don't think it would help the progress of the field to publish the article in the current form.

I tried to read that carefully and got lost after equation (4), the transition to equation (5) and (6) are not clear at all. I do not understand what is an approximation, what is a definition and what is a derivation.

Also (5) seems wrong in itself, the authors are trying to approximate a rectified linear network but it suggest that the activity will be equivalent to a linear network (at least when v(T) is small) ? And magically this changes in (6), and a clip non-linearity is introduced ?

The Figure 1 seems very encouraging at first, because it suggests that there is a clear and easy mapping between accumulating the spikes and computing a relu. I did not understand where this is appearing in the math and I cannot check whether the intuition conveyed by the figure is correct or not.

I was therefore hoping to see an empirical study of the difference between the SNN and the ANN: do the activity of the spiking neural network match the activity of artificial network? This is not shown.  I do not even understand if it is necessary to re-train the network to go back from the SNN to ANN or vice versa.

Since I had not understood the basics of the paper, it was impossible for me to understand the later section about the conversion error. My only take is that it seems wrong at first sight: how minimizing the error in the loss would minimize the mismatch between the network activity?

---

> ### Author Response · Authors · 2020-11-11
> **Response to Review # 3**
>
> We thank the reviewer for his/her effort in reviewing our paper. There are some definite misunderstanding of our paper that we want to clarify here. We guess that the main reason for R3 to feel lost in reading the context is that he/she may not be very familiar with the setup for SNN since the other two reviewers do not have the same problem in following the logic of our paper. Both of them are able to evaluate our paper with fair confidence and appreciate our contribution to both theory and practice. Based on R3's comments, we provide further explanations below and hope it can help R3 and other general readers to understand our paper.
>
> Let's first shortly describe the mechanism of the SNN. The SNN generates discrete spikes with respect to the series of events (this is why we usually have the simulation length parameter for training the SNN) while the ANN generates continuous values. Along the simulation process, the membrane potential accumulates on each SNN neuron and will release a post synaptic potential (PSP) to its linking neurons in the next layer when the accumulated membrane potential exceeds the threshold voltage. The activation values on SNNs are actually the spiking frequency multiplied by the threshold voltage.
>
> In terms of the conversion proposed in the current paper, from the perspective of statistical learning, a network, no matter it is SNN or ANN, is a parameterization of the sample space. Thus if two networks produce the same output for the same input, we would say the two networks are equivalent ignorant of the details in their infrastructures. And the goal of converting the ANN to SNN is to make the target SNN provide the most similar modeling with the source ANN on the sample distribution. This is an intuitive explanation of why we want to minimize the conversion error. For more rigorous discussion, one can refer to the information geometry theory that links the loss function and the metric on the space of distributions. In terms of the network activity, it is not our purpose to minimize the difference of layer-wise network activity between the source ANN and target SNN but the conclusion from the decomposition of conversion error that the overall conversion error can be minimized by approximating the activation values following a layer-wise strategy. This is why we do not perform a separate empirical study of the difference between the activity of the SNN and ANN but focus on the conversion error and the ultimate performance of the target SNN.
>
>
> For the transition from Eqn (4) to Eqn (5)(6), there is no magic but the transform of notations to illustrate the threshold balancing procedure between the source ANN and target SNN. As we state at the beginning of the previous paragraph, the SNN actually generates discrete spikes. Thus, if we want to compare the activation values between the source ANN and target SNN, we need to quantify the frequency of spikes in SNN averaged through the whole simulation sequence.  This is why we make the transition from Eqn (4) to Eqn (5). We guess the reviewer's confusion here is caused by the difference between $\mathbf{a}_{l+1}'$ and $\mathbf{a}_l'$. As we state in the context, the word “denote” means that $\mathbf{a}_l'$ is defined as the average input to the $l$-th layer. On the other hand, considering the network infrastructure of SNN, the $(l+1)$-th layer's input is also the PSP of the $l$-th layer. This relationship is described in Eqn (4) and transitioned to Eqn (5) in the average sense. For the transition from Eqn (5) to Eqn (6), remember that these $\mathbf{a}_l'$ s are discrete due to the features of SNN. In Eqn (5), the difference of the linear form is accumulated and absorbed in $\mathbf{v}^l (T)/T$. In Eqn (6), based on the discrete spiking mechanism, we calculate how much average PSP would be released as the input to the $(l+1)$-th layer when it accumulates $W_l\cdot\mathbf{a}_l'$ voltage on the $l$-th layer. Or more explicitly, the clip function can be viewed as a format of SNN’s activation function while the ANN takes ReLU in the same position. To say the least, as the reviewer also points out, this section forms the basics of the paper. If it was incorrect, there would have been systematic errors in the following approximation thus it would be impossible for us to maintain the SOTA results compared to existing works on the same topic.
>
>
> Regarding the reviewer's confusion in Fig 1, the panel A gives the overall schematics of the conversion pipeline and panel B and C talk about the intuition of layer-wise approximation in Sec 5. We will follow the reviewer’s suggestion and add a more explicit reference to avoid confusion here.
>
> Hope our clarification and explanation here can help solve the reviewer's concerns and facilitate the understanding of our paper for potential readers in general background. Any further comments are welcome as well.

---

> > ### Comment · AnonReviewer3 · 2020-11-16
> > **Still needs clarification**
> >
> > Considering the positive grades of the other two reviewers, I read the paper again more carefully. It is in fact better than I first thought! I apologize for the tone of my first review.
> >
> > I now understand that the value of this paper is to provide a theory that tells how one can reduce the conversion error by adding a threshold and a shift in the artificial model. The theory is fine (even though hard to read), but I wonder how much better is it in practice to the relu alone conversion from Dielh et al. 2015. If I read the performance figure and table correctly, it seems that the performance difference from a relu to SNN conversion is not extremely large and the number of time steps is not extremely larger without the threshold and the shift (all details equal otherwise). Is that correct?
> >
> > I find that the math is still unclear in a various points. I highly recommend adding a few clarifications. I am listing a couple of recommendations below and I will improve my grading if and once they are addressed. And even beyond the math, I still find it very unfortunate, that the paper lacks clarity in the writing. It really undermines the quality of interesting findings and it is a loss of time and energy for the reader.
> >
> > ## recommendations for modifications:
> >
> > - I now understand equation (4) to (6), what made be think it was fallacious was the second part of equation (1). What made a difference for me, was to understand that $\boldsymbol x_l'(t) = \boldsymbol \theta^{(l-1)}(t) $ (I may be helpful to add that in a comment). Therefore, $\boldsymbol x_l'(t)$ is the spike train, and writing $\boldsymbol x_l'(t) = h_j( W_l \cdot \boldsymbol x_{(l-1)}'(t) )$ in equation (1) is wrong because is $\boldsymbol x_l'(t)$ is the spike train and not the accumulated spike count $\boldsymbol a'_l$. The correct version of equation (1) is written later in equation (7), which I now understand. It would be great to correct equation (1) to make this less misleading.
> >
> > - I think the sentence: "We first derive the formula of converting the source ANN to target SNN with the threshold balancing
> > mechanism." is not clear for somebody who does not know what "threshold balancing" means. At least a reference to Diehl et al. 2015 in necessary here. I would also have appreciated a lame-man explanation of this term to anticipate what is going to be shown in the upcoming section. Something like: "Following the work of Diehl et al. 2015, we describe in this section the equivalence between a SNN and an ANN with thresholded relu non-linearity."
> >
> > - There are still obvious typos in the abstract, I think the word "filed" is meant to be "field". Also it is not clear what is meant here by "adaptive". I recommend that authors go again carefully through the entire paper to make sure that the writing is fine, at least according to them.

---

> > > ### Author Response · Authors · 2020-11-17
> > > **Further response to Review # 3**
> > >
> > > We really appreciate R3's open mind of re-evaluating our paper and helpful suggestions for improving our work's readability. We make modifications to our manuscript according to R3's advice and summarize our response below for ease of review.
> > >
> > > * The comparison with Dielh et al. 2015: On the MNIST dataset, we fit the same CNN as in Dielh et al. 2015 and applied our conversion with shift pipeline. Both SNNs used the Poisson spikes as the input. Our approach achieved an accuracy of 99.23~99.27% (randomness from the Poisson spikes) with 16 time points, while Dielh et al. 2015 achieved an accuracy of 99.12% with 0.5 seconds in 1000Hz, i.e., 500 time points. We also provided an illustrative example for ResNet20 on CIFAR-100 where we extend the simulation time to 2000 time points (Appendix A5, on page 14). We copy the text here.
> > >
> > >  "*In that case, although the conversion loss in both cases decays fairly fast, the SNN converted from the ANN with regular ReLU function converges to a plateau that suffers about 0.75% accuracy loss while the SNN converted from the ANN with threshold ReLU is almost loss-free. This result indicates that the improvement by adding threshold is not only on the converging efficiency but also on the final performance, which cannot be easily compensated through extending the simulation time in the original threshold balancing approach.*"
> > >
> > > * Clarification on Eqn (1): We thank the reviewer for providing the suggestion to avoid the confusion here. We change Eqn(1) as the reviewer suggested to represent the forward propagation on the average input values and average PSP to keep consistent with the following derivations and discussion. Please check the blue part in the revised manuscript from Page 2-3 for the details.
> > >
> > > * Explanation of the thresholding balance mechanism:  We add the reference and an explanation to the mentioned sentence. "*Following the threshold balancing mechanism \citep{Diehl2015FastclassifyingHS,Sengupta2018Going} that copies the weight from the source ANN to the target SNN, we here derive the forward propagation of PSP through layers in the target SNN.*"
> > >
> > > * Correction of typos and inaccurate word: We carefully recheck our manuscript to improve the grammar issues. Especially for the point R3 mentioned, we changed the word "adaptive" to "efficient" to fit better with the proposed method.
> > >
> > > We hope our revision addresses the reviewer's concerns and makes our manuscript easier to understand.

---

### Author Response · Authors · 2020-11-16
**General Response**

#### Summary
We appreciate the comments from all three reviewers on our paper. Some of them are concerned about the clearness of the proposed method. We want to make a brief clarification here about the mechanism of SNN and the value of our work. Detailed clarifications are included in the response to each reviewer separately.

The SNN generates discrete spikes with respect to the series of events (this is why we usually have the simulation length parameter for training the SNN) while the ANN generates continuous values. Along the simulation process, the membrane potential accumulates on each SNN neuron and will release a postsynaptic potential (PSP) to its linking neurons in the next layer when the accumulated membrane potential exceeds the threshold voltage. The activation values on SNNs are actually the spiking frequency multiplied by the threshold voltage. In this work, we propose a novel strategic pipeline that transfers the weights to the target SNN by combining threshold balance and soft-reset mechanisms. The method efficiently realizes the transfer with almost no accuracy loss. The theoretical analyses and comprehensive experiments can be viewed as the support of validity to each other.

We also modify the manuscript according to the comments and suggestions of the reviews. Specially, we make the following changes in the revised manuscript. The revised parts are colored in blue.

* We modify Eqn (1) to the version based on the average input.
* We explicitly add the definition of the threshold ReLU function on page 3.
* We summarize the conversion pipeline in Algorithm 1 on page 5.
* We provide an illustrative example of RNN in Appendix A.4 on page 13.
* We provide an illustrative example about the comparison of convergence for the SNN converted from the ANN with/without threshold in Appendix A.5 on page 14.

For details, please refer to the response to each review below.

---

### Decision · Program_Chairs · 2021-01-07
**Final Decision**

**Decision:**

Accept (Poster)

**Comment:**

The work tackles the task to convert an artificial neural networks (ANN) to a spiking neural network (SNN). The topic is potentially important for energy-efficient hardware implementations of neural networks. There is already quite some literature available on this topic.
Compared to these, the manuscript exhibits a number of strong contributions: It presents a theoretical analysis of the conversion error and consequently arrives at a principled way to reduce the conversion error. The authors test the performance of the conversion on a number of challenging data sets. Their method achieves excellent performances with reduced simulation time / latency (usually, in order to achieve comparable performance to ANNs, one needs to run the SNN for many simulated time steps- this simulation time is reduced by their model).
One reviewer criticized that the article was hard to read, but this opinion was not shared by other reviewers and the authors have improved the readability in a revision.

In summary, I believe that this manuscript presents a very good contribution to the field.